# Effective Components of Behavioural Interventions Aiming to Reduce Injury within the Workplace: A Systematic Review

**Mairi Bowdler \*, Wouter Martinus Petrus Steijn**  **and Dolf van der Beek**

Netherlands Organisation for Applied Scientific Research TNO, Sylviusweg 71, 2333 BE Leiden, The Netherlands; wouter.steijn@tno.nl (W.M.P.S.); dolf.vanderbeek@tno.nl (D.v.d.B.)
\* Correspondence: mairi.bowdler@tno.nl

**Abstract:** For years, the connection between safety behaviours and injury and illness in high-risk industries has been recognised, but the effectiveness of this link has been somewhat overlooked. Since there is still a significant amount of injury within high-risk workplaces, this systematic review aims to examine the effectiveness of behavioural interventions to decrease fatal and non-fatal injuries within high-risk industries. Scopus and Google Scholar were used to find relevant systematic reviews and meta-analyses on this topic. In total, 19 articles met the inclusion criteria. Of these articles, 11 suggested that their reviewed interventions revealed some evidence of being effective in reducing injury/accident rates. Additionally, seven of the papers found that the interventions affected certain determinants, such as safety knowledge, health and safety behaviours, attitudes, efficacy, and beliefs. One of the papers found no effect at all. It must be noted that a significant amount of the articles ($n = 10$) reported methodological quality or quantity issues, implying that the results should be approached with caution. Nonetheless, it was found that certain components, such as multi-faceted interventions tailored to the target group, contribute to either reducing injury/accident rates or improving the specific aforementioned determinants. There is a need for additional safety interventions in high-risk industries that are based on methodologically sound structural elements and theoretical frameworks. Existing approaches, such as Intervention Mapping, can assist safety professionals in achieving this goal.

**Keywords:** systematic review; meta-review; occupational safety; fatal and non-fatal injury; behaviour; intervention

## 1. Introduction

For decades, organisations such as the National Safety Council (USA) have been aware of an association between human behaviour and injury and illnesses. Behaviour has been addressed as an essential origin of injury and illness [1], although rarely as the single contributing factor. Research from the last century within the safety domain has shown that incidents commonly occur as part of an intricate mixture of factors that come into play, like latent failures in the organisational system stemming from, for instance, a badly implemented safety management system (barrier management), not adhering to legislative requirements, a weak safety climate or culture, and/or lacking proper engineering controls or designs [2]. Nevertheless, it is recognised that insufficient attentiveness towards safety behaviours from employers of high-risk workplaces (e.g., construction, manufacturing, agriculture) has been acknowledged as a major cause of occupational fatal and non-fatal injury in various studies across the globe, e.g., UK [3], Netherlands [4], Japan [5], USA [6] and Taiwan [7]. This underlines the need to focus on modifying or redirecting processes associated with behaviour by targeting unsafe or risky behaviours as a promising method to reduce injury rates [8]. The importance of this requirement is underscored by the persistently high occurrence of both fatal and non-fatal injuries in high-risk work environments; according to reports, approximately 50% of severe workplace accidents and

the majority of fatal accidents are attributed to high-risk injuries [9]. In 2018, within Europe (EU-27), industries considered high-risk made up 66% of the 3332 fatal injuries within the workplace [10].

Safety interventions encompass deliberate measures aimed at fostering safety and reducing the occurrence or severity of workplace accidents [11]. Behavioural safety interventions, in particular, try to change safety-related behaviours in a direct manner with behavioural principles and various strategies. This meta-review considers multiple strategies (e.g., peer observation, incentives, feedback, and specific safety training) to achieve behaviour modification and then reduce incidents as the intended goal [12]. We consider training as a strategy for this article since we regard behavioural interventions as a generic approach.

Interventions targeting safety behaviours can contain specific key components that improve their effectiveness in reducing incidents. For example, interventions that specifically target behaviours are recognised as having an increased positive effect if they are based on theoretical apprehensions [13–15] since theory can aid the identification of cognitive, motivational, and emotional states that trigger certain behaviours. These findings have also been observed in health-related behaviours [16–18]. Therefore, this identification helps determine which specific factors influencing behaviour should be focused on by interventions. This implies that targeting the modification of certain behavioural determinants is expected to result in a change in behaviour [19]. Behaviour Based Safety (BBS) is an intervention which views the main cause of injuries as being due to unsafe behaviour [20] and is related back to Skinner's operant conditioning [21]. Thus far, BBS has yielded some positive findings with regard to incident levels and safety performance within high-risk workplaces [22,23]. Another example of a key component that could improve effectiveness is interventions that apply a multi-faceted approach in which personal attributes of the employee, such as physical and/or mental health, are targeted in combination with safety behaviours [24–26]. This type of intervention often appears under the label of 'Occupational Health and Safety'/'Occupational Safety and Health training' (OHS/OSH). These interventions are administered as a means of incorporating occupational safety and injury prevention along with health promotion to protect and promote employee health, safety, and well-being [24]. Research suggests there are notable positive effects on determinants such as worker engagement when personal health behaviour interventions are implemented in combination with occupational safety [24].

Based on the authors understanding, there is still a dearth of research to date that specifically aims to ascertain the fundamental elements that render safety interventions effective. The investigation into specific determinants appears to be both constrained in scope and yields contradictory findings. For example, while occupational safety literature often highlights the importance of safety knowledge, the available evidence on this matter can be seen as conflicting. Safety knowledge is described as a "proximal antecedent" [27] (p. 1104) of safety behaviours since it supplies employees with certain assets to know how to perform safely [28]. In contrast, Nahrgang, Morgeson, and Hofmann [29] discovered that, in fact, it was safety climate rather than safety knowledge that explained a larger amount of variance in safety behaviours. Additionally, a recent study by Fabiano et al. [30] claimed that the behaviour of workers is said to be influenced by considering four pertinent categories, namely behaviour, attitude towards safety, response to near-miss/incidents, and communication.

A clear overview of key components for safety behaviour interventions will be useful for the overall understanding of the processes of improving safety within this field, especially within high-risk industries where there is a heightened risk of accident/injury [9,10]. To the best of the author's knowledge, this represents the inaugural review of review papers focusing on occupational safety. Therefore, this information will also have practical implications for the future development of new interventions. It will improve the current situation regarding overall evidence on safety interventions, which is currently considered insufficient in design or generally limited [12].

Our aim is to evaluate published systematic reviews and meta-analyses on interventions targeting safety behaviours to identify key effective components to reduce fatal and non-fatal injury in high-risk industries. In effect, we will provide an overview of the currently available literature and pose the following research questions:

- What is the current knowledge on the efficacy of occupational safety interventions focused on behavioural change in reducing fatal and non-fatal injury?
- Which key components can be identified of these interventions that make them more effective in reducing fatal and non-fatal injury?

## 2. Materials and Methods

This paper is referred to as a systematic review which conducted an analysis of published systematic reviews and meta-analyses evaluating behavioural interventions aiming to reduce fatal and non-fatal injury in the workplace. Henceforth, we will refer to our systematic review as a meta-review and refer to the reviewed papers as systematic reviews and meta-analyses accordingly. See Figure 1 for the steps taken for the data collection. Scopus and Google Scholar were utilised as the search tools for the articles reviewed in this paper. Scopus is an abstract and citation database published by Elsevier, including peer-reviewed papers from, among others, the disciplines of life sciences, social sciences, physical sciences, and health sciences. Scopus is Elsevier's abstract and citation database covering over 35,000 (peer-reviewed) publications from life sciences, social sciences, physical sciences, and health sciences. Google Scholar was used to supplement our results with papers that may not have been found in Scopus. Keyword search terms included the following: (occupational OR workplace) AND safety AND intervention AND ("meta-analysis" OR "systematic review") AND (behaviour OR behaviour) AND effective* (The asterisk indicates that any characters could follow, allowing the search to also include 'effectiveness'); and (occupational OR workplace) AND intervention AND (accident OR casualty OR injury) AND ("meta-analysis" OR "systematic review") AND (behaviour OR behaviour) AND effective*. The search was completed in the months of September–November 2020.

The initial search resulted in 93 papers, from which 15 were omitted due to being duplicates. From the remaining 78 papers, another 59 papers were omitted based on a review of their abstracts and titles for eligibility on the foundation of the following criteria:

- Documentation in the English language;
- Systematic reviews/meta-analyses;
- Assessment of (safety) behavioural interventions;
- Aim to reduce fatal and non-fatal injury;
- Target group of 'high-risk' occupations (e.g., employees in construction, agriculture, manufacturing industries).

The remaining 19 papers were made up of various study designs such as randomised control trials (RCTs), quasi-experimental trials, controlled pre-post studies and interrupted time series and cross-sectional studies. The publication date of the reviewed papers ranged from the year 2000 to 2018, and the publication date of the intervention studies within the reviews ranged from 1966 to 2017. The target population consisted of adult (aged >18 years) employees, and the sample sizes of the intervention studies ranged from micro to large.

Two of the authors assessed the risk of bias in all the included reviews independently using the ROBIS tool [31]. See Tables 1 and 2 for the final rating of each of the reviewed papers (weak, mediocre, good). The ROBIS tool was developed specifically for systematic reviews using thorough methodology and particularly focuses on broad categories of reviews within healthcare settings, including interventions. The tool undergoes three phases of completion: (1) optional assessment of relevance, (2) identification of concerns regarding the review process, and (3) evaluation of the risk of bias in the review. Signal questions are incorporated to aid in assessing concerns related to potential biases in the review. The ratings derived from these signal questions assist assessors in making an overall judgment of the bias risk. Two authors of this paper utilised the tool from phase 2; to identify concerns with the review process, then completed phase 3, in which the

overall risk of bias is assessed. Any discrepancies between the two assessors were resolved through discussion.

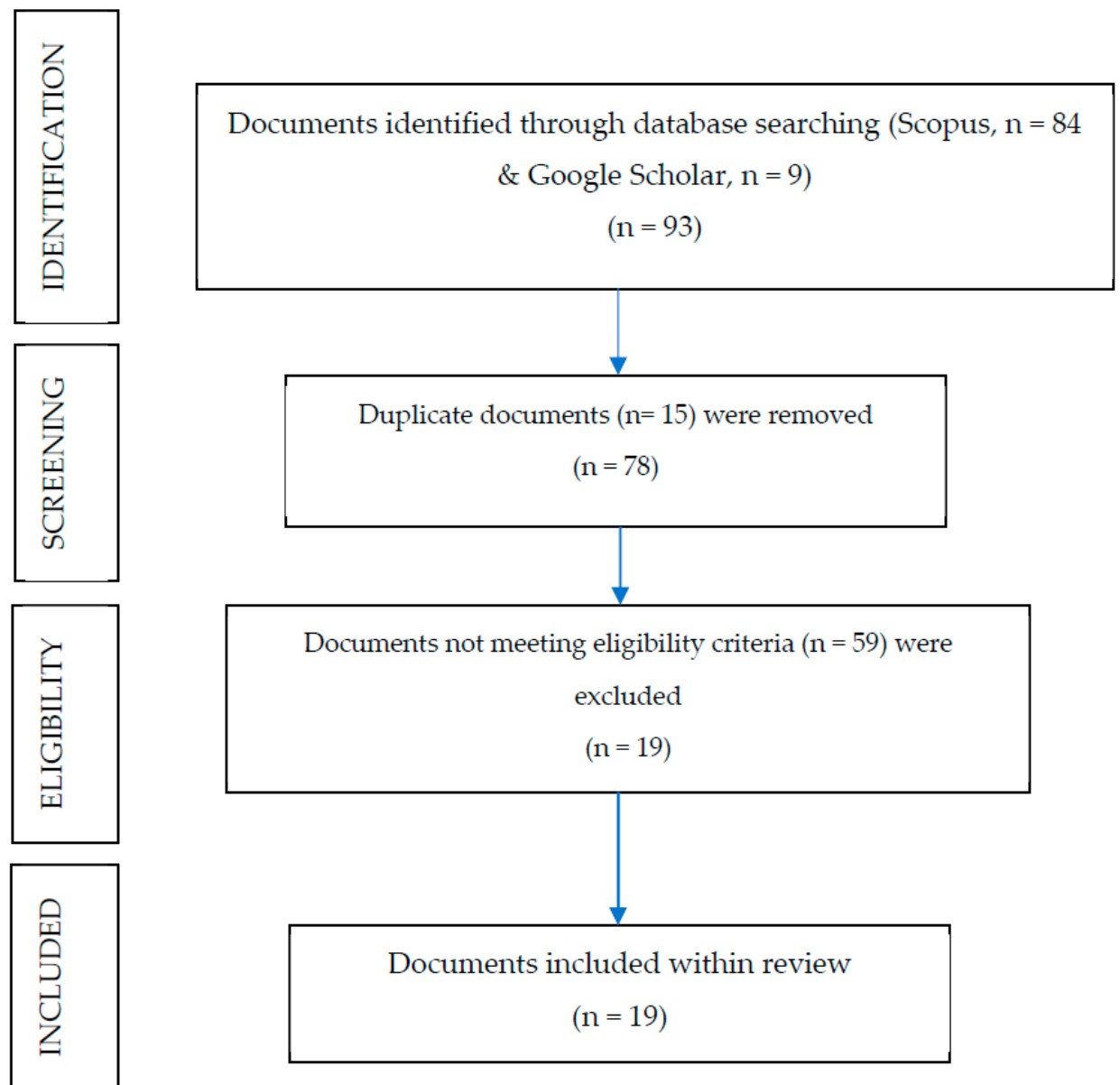

**Figure 1.** Flow chart of study sample process.

**Table 1.** Summary of results on studied characteristics.

| | ROBIS Score | | | Components | | | Meth Quality |
|---|---|---|---|---|---|---|---|
| | Good | Med | Weak | Multi-faceted | Tailor | Poor | Suff/Not Mentioned |
| Effect on injury reduction (and determinants) | 11 | 7 | 0 | 4 | 5 | 2 | 7 | 4 |
| Effect only on determinants | 7 | 4 | 2 | 1 | 4 | 3 | 3 | 4 |
| No effect | 1 | 1 | 0 | 0 | 1 | 0 | 0 | 1 |

Table 2. Complete overview of the studies assessed within reviews.

| Reference | Goal | Studies | Design | Intervention | Industry | Sample Size (Participant No.) | Countries | Effectiveness | Risk of Bias (ROBIS) |
|---|---|---|---|---|---|---|---|---|---|
| [24] | To investigate the effectiveness of integrated approaches | 31 | 14× Experimental trials with randomized group assignment 9× Quasi-experimental trials 8× Single group design with pre-post comparisons | Occupational safety and health promotion—Multi-faceted [integrated] approach | 11× Manufacturing 12× Health care 3× Construction 2× Fire services 1× Telecommunication services 2× Other | 6× Medium 1× Small 24× Large | 2× Canada 17× USA 1× Japan 11× Europe | Yes, for interventions targeting employee physical and mental health | Good |
| [32] | To evaluate the effectiveness of interventions to prevent occupational injury associated with construction work | 5 | 4× ITS 1× Controlled ITS study | 1× Multifaceted safety campaign 1× multifaceted drug-free-workplace program 3× injury-reducing effect of legislation | Construction | 5× Large | 3× USA 1× Denmark 1× unknown | Yes, limited evidence was found for reducing the level and the trend of injuries. Inadequate study designs of review papers noted. | Good |
| [33] | To evaluate the effectiveness of interventions designed to prevent work-related eye injuries | 7 | 6× Before and after comparison 1× Case–control type analysis | Vision screening, education, provision of glasses, policy change 2× primary behavioural interventions—a multi-faceted approach | 2× Shipping industry 5× Manufacturing | Not reported—small numbers assumed | Not reported | Yes, limited evidence that policy changes are effective at reducing eye injuries. Inadequate study designs of review papers noted. | Weak |
| [34] | Assess the effects of occupational safety and health regulation enforcement tools for preventing occupational diseases and injuries | 23 | 2× RCTs 2× CBAs (respective cohort studies or quasi-experimental studies) 1× ITS 12× Panel studies 6× Qualitative studies | OSH regulation enforcement interventions—tailored approach | 7× Manufacturing 2× Construction 1× Woodwork 10× Workplace with high amount of physical work 3× Other | 2× Small 21× Large | 4× Canada 16× USA 1× Sweden 1× South Africa 1× Australia | Yes, weak evidence that inspections decrease injury. Inadequate study designs of review papers noted | Good |
| [35] | Review the evidence for the effectiveness of active behaviour change safety interventions in the construction industry | 15 | 4× RCT 1× Four-group Solomon design 4× Pre-post 3× Interrupted time-series 2× Mixed approach incl. pre-post and time series 1× Within-group design | Interventions used a range of methods to change behaviour, including coaching, educational/information sessions, and computer games. | Construction | 5× Large 2× Medium 8xUnknown | 6× USA 7× Europe 1× Hong Kong 1× India | Yes, inconsistent evidence that interventions improve injury rates. More consistent evidence of improvement in safety behaviour. Inadequate study designs of review papers noted. | Good |

**Table 2.** *Cont.*

| Reference | Goal | Studies | Design | Intervention | Industry | Sample Size (Participant No.) | Countries | Effectiveness | Risk of Bias (ROBIS) |
|---|---|---|---|---|---|---|---|---|---|
| [36] | To review evidence concerning the effectiveness of workplace drug testing as a workplace safety strategy | 23 (17 tested reduction in injury rates) | 7× Time series design 7× Cross-sectional 2× Pre-post designs 1× Matched-pairs design | Workplace drug testing | 2× Manufacturing 5× Transport 3× Construction 2× Retail 5× Other | 3× Medium 2× Small 4× Large 14× Unknown | of the 17/23: 16× USA 1× Unknown | Yes, inconsistent evidence that drug testing is associated with a reduction in accidents. Inadequate study designs of review papers noted. | Good |
| [37] | Assess the effectiveness of interventions aiming to prevent occupational injury among workers in the agricultural industry | 8 | 3× RCTs 2× cRCT 3× ITS | 3× Multi-faceted approach with educational interventions incl: (non-)OSH professionals, written info, and financial incentives | Agriculture | 1× Large 7× Unknown | 3× USA 4× Europe 1× Sri Lanka | Yes, weak evidence that financial interventions could be effective in reducing injury rates. Educational interventions are not effective (as stand-alone). | Good |
| [38] | Review the evidence for the effectiveness of different strategies to prevent falls from heights in the construction industry | 3 | 2× Before and after comparison 1× Company comparison | Various: environmental modifications, educational, administrative, and legislative. | 2× Construction 1× Ship work | Not reported | 1× USA 1× Finland 1× Hong Kong | Yes, weak evidence that regulations might decrease fall injury rates. Inadequate study designs of review papers noted. | Weak |
| [39] | To estimate the summary effectiveness of different needle-stick injury (NSI)-prevention interventions | 17 | 1× RCT 16× Before-after comparisons | Training, safety-engineered devices, or the combination of training and SEDs—a multi-faceted approach | Healthcare | Not reported | 5× USA 8× Europe 1× Iran 1× Australia 1× Pakistan 1× Saudi Arabia | Yes, the intervention reduced the risk of injury. | Weak |
| [40] | To assess the effectiveness of behaviour-based safety (BBS) interventions in reducing accidents and injury occurrence in occupational settings | 13 | 1× Study with control group 12× No control group | Stand-alone or a combination of safety training, feedback, goal setting, token economy and poster campaigns were the main study variables to reduce accidents/injuries. | 7× Manufacturing 2× Shipyard or marine engineering 4× Other | 2× Large 2× Medium—large 5× Medium 3× Small 1xUnknown | Not reported | Yes, evidence for a significant reduction in injuries/accidents after BBS intervention. Inadequate study designs of review papers noted. | Good |

**Table 2.** *Cont.*

| Reference | Goal | Studies | Design | Intervention | Industry | Sample Size (Participant No.) | Countries | Effectiveness | Risk of Bias (ROBIS) |
|---|---|---|---|---|---|---|---|---|---|
| [41] | Assess the effectiveness of Joint Health and Safety committees and how to make them effective | 31 | 25× Cross-sectional studies 6× Review studies | Joint Health and Safety committees—tailored approach | Industrial companies from various sectors: metal, plastic, grain, textile | Not reported | 15× Canada 4× USA 2× Australia 3× Europe 7× unknown | Yes, effective implementation of safety committees results in a safer workplace | Weak |
| [42] | To identify and synthesise the literature results about the effectiveness of OSH training programs for migrant workers in the agricultural sector | 29 | 9× Cross-section studies 20× Within-subject experimental studies | OSH training—Tailored intervention design | Agriculture (Migrant farm workers) | 12× Large 13× Medium 3× Small 1× Micro | 28× USA 1× Australia-Indonesia | No effectiveness was reported in reducing health outcomes. Weak effect reported on improving safety knowledge, safety behaviours and safety attitudes and beliefs. | Mediocre |
| [43] | To investigate evidence on the effectiveness of farm injury prevention interventions | 25 | 9× Post-tests 9× Pre-post-tests 1× RCT 1× Questionnaire 2× Ongoing surveillance 3× Not reported | 5× multifaceted interventions 9× farm safety programs without completed evaluations 11× education programs | Agriculture (Migrant farm workers) | Not reported | 18× USA, 4× Europe 1× Australia 2× not reported | No, but weak evidence on improving efficacy. Inadequate study designs of review papers noted. | Mediocre |
| [44] | To evaluate the effectiveness of interventions that aim to enhance the use of hearing protection | 7 | 1× Randomised trial 3× Randomised control trial 2× Controlled trial randomised by clusters 1× Randomised experimental design | Educational, behavioural, and technical—mix multi-faceted and tailored approaches | Students, engineers, labourers, and young people working on a farm are exposed to noise levels above 80dB | 7× Large | 7× USA | No, but limited evidence in promoting safety behaviour. Inadequate study designs of review papers noted. | Good |
| [45] | To evaluate evidence on the benefits and harms of integrated Total Worker Health interventions | 15 | 12× RCT 2× nRCT 1× Prospective cohort study | Integrated Total Worker Health Interventions—Multi-faceted approach | 7× Manufacturing and construction 4× Health care and social assistance industry 4× Other | 11× Large 2× Medium 2× Small | 9× USA 6× Europe | No, but some effectiveness in improving health behaviour. | Good |

**Table 2.** *Cont.*

| Reference | Goal | Studies | Design | Intervention | Industry | Sample Size (Participant No.) | Countries | Effectiveness | Risk of Bias (ROBIS) |
|---|---|---|---|---|---|---|---|---|---|
| [46] | Assess the effectiveness of sun-safety education programmes in outdoor occupational settings and an overview of outdoor workers' sun-related knowledge, attitudes, and protective behaviour | 52 (34 relevant articles, 18× interventional studies) | Of the 18× interventional studies: 1× Non-randomised 10× Randomised 2× Cross-sectional 5× Pre-post test | Educational programmes | 15× Agricultural workers/farmers 13× Construction/road workers 7× Aquatic personnel | 9× Large 7× Medium 1× Small 1× Unknown | 27× USA 11× Europe 10× Australia/New Zealand 2× Israel, 1× Brazil 1× Japan | No, but occupational sun-safety education is effective in improving safety behaviour. | Good |
| [47] | To verify the efficacy of occupational health and safety (OHS) training in terms of knowledge, attitude, beliefs, behaviour, and health. | 28 | 21× RCT and quasi-experimental studies 7× Not reported | Classroom theory lessons with various active teaching—tailored approaches | 7× Construction 6× Agriculture 5× Healthcare 4× Tertiary 3× Manufacturing 3× Other | 3× Large 12× Medium 8× Small 2× Micro 3× Unknown | 9× Europe 8× USA 1× Taiwan 1× Israel 1× India 1× Brazil 7× unknown | No, but training is effective at improving attitudes and, beliefs, knowledge, but less so at improving behaviour | Weak |
| [48] | To assess whether OSH training has a beneficial effect on workers | 22 | 22× RCT | Variety of training interventions—a multi-faceted approach | 6× Healthcare 6× Office workers 2× Agriculture 2× Construction 1× Miners 5× Other | 12× Large 9× Medium 1× Small | 11× USA 8× Europe 2× Canada 1× China | No, but education does affect behaviour. Inadequate study designs of review papers noted. | Good |
| [49] | To assess the effects of interventions for preventing injuries in construction workers | 17 | 14× Interrupted time series 3× Controlled before-after studies | 3× Multi-faceted 10× Compulsory 3× Educational 1× Facilitative | Construction | 6× Unknown 11× Large | 6× USA 11× Europe | No, no evidence for or against effective interventions for reducing injuries | Good |

### 3. Results

Please refer to Table 1 for a summary of the primary findings from the review papers pertaining to the characteristics under investigation.

The final ROBIS ratings revealed that the overall risk of bias from the reviewed papers themselves was quite low; twelve were concluded as 'good', two as 'mediocre', and five as 'weak'. The weak scores were a result of lacking information concerning their data collection, making the overall quality difficult to judge.

In total, 11 of the 19 papers [24,32–41] found that the assessed interventions influenced the reduction of injury/accident rates. From the ROBIS analysis, seven of these eleven review papers were concluded as "good" and four as "weak". Tuncel et al. [40] wrote one of the review papers that gained a "good" score from the ROBIS assessment as well as finding evidence for a significant reduction in injuries/accidents after the application of a BBS intervention. On the other hand, although Tarigan et al. [39] found their assessed needle-stick injury (NSI)-prevention interventions also reduced the risk of injury, our ROBIS score concluded this review paper as "weak", suggesting their conclusions should be approached with caution.

Additionally, seven of the papers [42–48] found that the interventions had an effect on certain determinants, such as safety knowledge, health or safety behaviours, attitudes, efficacy, and beliefs. For instance, Ricci et al. [47] found that OSH training had a positive influence on workers' attitudes and beliefs, with a slightly lesser impact on their knowledge as well. Similarly, both Reinau et al. [46] and Robson et al. [48] found no effect on injury rates, but the assessed interventions within their reviews both showed an effect on behaviours. Of these seven papers, it is difficult to claim a causal link between these determinants and injury rates, due to the poor methodological designs of the interventions assessed.

One of the papers [49] examined interventions aimed at preventing injuries in construction workers and found no evidence of effectiveness in reducing injury rates or influencing specific determinants.

From the analysis, certain components were mentioned to contribute to the effectiveness of either reducing injury/accident rates or improving the certain aforementioned determinants. Ten of the nineteen papers [24,32,33,37,39,43–45,48,49] concluded that a multi-faceted approach improves the effectiveness of the intervention. These interventions applied an approach that involved using multiple facets or components, each designed to target different aspects of the problem or issue at hand. For example, Lehtola et al. [32] assessed interventions, including a multi-faceted safety campaign and a multi-faceted drug-free workplace program, and both interventions found evidence that they reduce (non)fatal injury. Furthermore, Rautiainen et al. [37] found that educational interventions alone were not effective in reducing injury rates.

Five of the 19 papers [41,42,44,45,47] concluded that an intervention that has been tailored to the target group improves the effectiveness of the intervention. This type of intervention is customed or individualised to meet specific characteristics of the individual/target group, leading to heightened engagement. For example, El Dib et al. [44] found that the interventions utilising a tailored design resulted in higher mean usage of hearing protective devices as compared to no intervention.

One of the main results concerns the methodological design of the intervention studies within the reviewed papers. Ten of the 19 papers [32–36,38,40,43,44,48] concluded that their review evidence was weak due to either the quality or quantity of the evidence. Please refer to Table 2 for a comprehensive summary of the reviews analyzed in this meta-review, encompassing all findings and additional details derived from this analysis (e.g., the number and type of interventions evaluated and the target groups associated with these interventions).

## 4. Discussion

This meta-review set out to establish current knowledge from systematic reviews and meta-analyses on the efficacy of behavioural safety interventions in high-risk workplaces and make an attempt at identifying key components that may contribute to their effectiveness.

Of the reviews and meta-analyses included in this paper, the majority concluded that the interventions reviewed were effective in either reducing injury rates and/or positively affecting another determinant, e.g., safety behaviour. However, a significant portion of them had poor methodological quality and/or quantity. It has been previously stated that a large amount of evidence concerning the efficacy of occupational safety is of poor quality [12,50], and this reoccurring deficiency across the literature makes it difficult to evaluate and compare studies.

We expose here the need for research to be carried out within this field that utilises experimental/quasi-experimental designs (e.g., RCTs). These designs have heightened credence [48] and are recognised as a reference standard for understanding intervention causal relationships [51]. Additionally, we would suggest follow-ups and longer durations (e.g., one–four years, [35]) of intervention implementation, as a longer duration of the observed effects provide more credible validity for the effectiveness of the intervention in terms of sustained behaviour change over time. Moreover, Mischke et al. [34] noted that long-term commitment from the organisation is necessary to improve both health and safety within the workplace.

Despite the methodological issues described above, by comparing the papers, we were still able to draw some meaningful conclusions concerning key components that could aid the effectiveness of these interventions.

The results showed some of the interventions were capable of impacting injury/accident rates; other interventions had an effect on certain determinants such as behaviour, knowledge, attitudes, efficacy, and beliefs [42–48]. Although these determinants were improved, this does not imply a decrease in injury/accident rates. In terms of safety knowledge, previous literature has reported mixed findings on its ability to reduce injury/accident rates; some claim increasing knowledge has been claimed as the first stage of embracing new ideas and a major contribution to changing behaviour, and in effect, reducing fatal and non-fatal injury [52]. Still, others state that lacking safety knowledge can lead to increased injuries and safety errors [53–55]. Based on the analysis presented in this paper, there seems to be no evident association between knowledge and injury/accident rates. However, this is primarily attributed to the insufficient study design rather than the determinant itself. The same conclusion applies to the other determinants mentioned earlier, such as attitudes and beliefs.

Research conducted in the previous century within the field of safety has demonstrated that incidents often arise because of a complex combination of factors [2]. Additionally, since human behaviour is highly dynamic and multifactorial in origin [56], it seems plausible that a multifaced approach was said to have effective qualities in the reviewed papers (e.g., [48,49]). Interventions targeting more than one determinant have been claimed as being more effective than other techniques targeting singular components [34,49]. There are promising findings emerging from the combination of targeting both employee health and safety, such as heightening worker engagement [24]. Additionally, for decades research has acknowledged that health issues such as stress could contribute to an increase in workplace fatal and non-fatal injuries [57].

Interventions that were tailored to the individuals from the target group also appeared to be an effective component of the interventions. Feltner et al. [45] claimed that the most effective interventions they assessed were those that were tailored to the cultural or social components of the worker population. The tailored interventions also appeared to influence worker engagement. This finding is encouraging as it aligns with previous research and supports Caffaro et al.'s [42] assertion that greater engagement in interventions leads to more significant positive effects.

Based on our results, we suggest future research within this area should be methodologically sound and clearly identify and describe numerous key components that may be contributing to the effectiveness of reducing accidents/injury within high-risk workplaces. Additionally, we recommend multi-faceted approaches, where several determinants at different levels (e.g., technical, organisational, behavioural, social) are targeted, as well as interventions that have a tailored design that is unique to the target group at hand.

To achieve these goals, researchers could investigate the Intervention Mapping approach [57,58]. The Intervention Mapping approach is a well-documented method to aid in the development of interventions to ensure that they address the needs of the target population. After establishing the needs and change objectives, Intervention Mapping guides the developer into establishing various program components based on theory to ensure that the change objectives are affected. The method involves six steps: conducting a needs assessment, setting clear objectives, selecting theory-based methods and strategies, developing intervention components, planning for adoption and implementation, and evaluating the intervention's effectiveness.

Certain limitations need to be considered when considering the results presented. Due to the absence of adequate methodological designs and comprehensive descriptions in the observed reviews, we have been unable to provide a comprehensive empirical summary of workplace interventions aimed at enhancing safety.

Furthermore, the reviews included in this meta-review evaluated enterprises of varying sizes and encompassed a wide range of industries. As a result, meaningful comparisons were not feasible, preventing us from drawing conclusions regarding the effects of interventions based on enterprise size or type. We recommend that future studies on this topic make a clear note of the size and type of industry to assess whether there is any link to the effectiveness of these interventions. We would, however, like to note that construction was the most common industry, followed by maintenance.

Lastly, despite the limited number of eligible reviews, we employed the ROBIS tool to evaluate the risk of bias in each of the reviews included in this paper. The results indicated that the overall risk of bias in the reviewed papers was relatively low, as detailed in the Section 3. Additionally, it is important to mention that we did not utilise the specialised health database 'Medline' for the reasons mentioned earlier. Our focus was primarily on the field of occupational safety and health, whereas Medline primarily concentrates on other fields, such as patient safety. We understand the limitations and biases that may be partnered with these decisions. We also recognise that the search terms applied to the databases included within this review also highlight another limitation of this study since they can cause certain biases. For example, reviewing studies published only in the English language introduces language biases.

## 5. Conclusions

Our meta-review uncovered that interventions aiming to improve safety behaviours in high-risk workplaces can be effective. However, an abundance of this body of work is undetermined due to poor methodological design and/or limited data. Hence, we strongly advocate for further research that adheres to rigorous and methodologically sound practices to ascertain the most effective behavioural approaches for enhancing safety in high-risk industries. Our work found that some of the behavioural interventions assessed within the reviews were able to cause a reduction in injury/accident rates. We also found that some interventions had key components that contributed to their effectiveness. Specifically, interventions that are multi-faceted in nature, i.e., combining different methods to affect individuals and interventions that are tailored to their target population, are more likely to be effective. We suggest the adoption of Intervention Mapping in future studies, as it provides researchers with a structured framework for designing evidence-based interventions that consider the needs and preferences of the target population. In conclusion, we stress the importance of conducting a substantial amount of methodologically rigorous research

to identify the most effective behavioural approaches for enhancing safety in high-risk industries.

**Author Contributions:** Conceptualisation, M.B., W.M.P.S. and D.v.d.B.; methodology, M.B.; formal analysis, M.B. and W.M.P.S.; writing—original draft preparation, M.B.; writing—review and editing, M.B., W.M.P.S. and D.v.d.B. All authors have read and agreed to the published version of the manuscript.

**Funding:** This research received no external funding.

**Institutional Review Board Statement:** Not applicable.

**Informed Consent Statement:** Not applicable.

**Data Availability Statement:** The data presented in this study are available in Tables 1 and 2.

**Conflicts of Interest:** The authors declare no conflict of interest.

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
