# Peer review of "Effective Components of Behavioural Interventions Aiming to Reduce Injury within the Workplace: A Systematic Review"

_safety, 2021_

Round 1
Reviewer 1 Report
The authors proposed a systematic review to investigate the effectiveness of behavioral interventions aiming to decrease fatal and non-fatal injury within high-risk occupational industries Some changes are advised. - The abstract should be improved, clearly stating the problem, establishing the objectives of the research, defining the methodology, as well as giving details of results and the main conclusions obtained; - In the Introduction section the authors need to point out how this study is different from other ones. My suggestion is to deeply think about two things: why a new review paper is needed in this topic, and what is the specific novel result for their study. The introduction should be written in a way that leads the readers understand these two points; - Please, add at least 10 references to support your points for example cosidering human error probability: A) Quality Checks Logit Human Reliability (LHR): A New Model to Evaluate Human Error Probability (HEP) Di Bona, G. , Falcone, D., Forcina, A., De Carlo, F., Silvestri, L.Mathematical Problems in Engineering, 2021, 2021, 6653811 B) Systematic Human Reliability Analysis (SHRA): A New Approach to Evaluate Human Error Probability (HEP) in a Nuclear Plant Bona, G.D. , Falcone, D., Forcina, A., Silvestri, L. International Journal of Mathematical, Engineering and Management Sciences, 2021, C) Hybrid fuzzy AHP-TOPSIS framework on human error factor analysis: Implications to developing optimal maintenance management system in the SMEs Velmurugan, K. , Saravanasankar, S. , Venkumar, P., Sudhakarapandian, R. , Bona, G.D. Sustainable Futures, 2022, 4, 100087
D) Integrated hazards method (IHM): A new safety allocation technique
Falcone, D., Di Bona, G., Duraccio, V., Silvestri, A. ,- Consider adding a block diagram for better explaining the methodology; - Please, highlight limitations of your work in the Conclusion Section; - Please elaborate on the future research needs in this domain;
Good
Author Response
Manuscript re-submission Cover letter
Leiden, 20-06-2023
Dear Reviewer 1,
We would hereby like to resubmit the manuscript under the title “Effective components of behavioural interventions aiming to reduce injury within the workplace: a systematic-review.” We would like to thank the reviewer for providing us with the reviewer comments that have helped us improve on the manuscript. The exact nature of our changes and our response to the reviewer comments have been submitted elsewhere (please see the attachment).
We confirm that neither the manuscript nor any parts of its content are currently under consideration or published in another journal. All authors have approved the manuscript and agree with its submission to MDPI Safety.
We look forward to hearing from you.
Sincerely,
Mairi Bowdler, Wouter Steijn & Dolf van der Beek
Mairi Bowdler (corresponding author)
TNO
Sylviusweg 71
2333 BE Leiden
e-mail: mairi.bowdler@tno.nl

Reviewer 2 Report
The paper describes research to improve the effectiveness of interventions to reduce the frequency of injury at the workplace.
The structure of the paper is fine. The discussion is clear, and the conclusions follow the findings and present recommendations. Section 3 Results should be presented in a way more digestible for a reader, e.g., by adding a few graphs.
Rightfully, in Section 4 Discussion it is stated that the conditions at the workplace should obtain more attention when doing and reporting a study. It would certainly be recommended to perform ahead of the intervention analysis a measurement of the safety climate. This is because if management is not safety minded and does not support the intervention activity, the effectiveness improvement of the employees by an intervention will easily be lost.
On line 119 the ROBIS tool is mentioned. Because the tool is a key classification instrument in the study, it is recommended to spend a few lines explaining the principles of the tool. The same holds for Intervention Mapping (line 235). Also, please check the reference numbering, as ROBIS is referenced in [23] and not in [24].
Author Response
Manuscript re-submission Coverletter
Leiden, 20-06-2023
Dear Reviewer 2,
We would hereby like to resubmit the manuscript under the title “Effective components of behavioural interventions aiming to reduce injury within the workplace: a systematic-review.” We would like to thank the reviewer for providing us with the reviewer comments that have helped us improve on the manuscript. The exact nature of our changes and our response to the reviewer comments have been submitted elsewhere (please see the attachment).
We confirm that neither the manuscript nor any parts of its content are currently under consideration or published in another journal. All authors have approved the manuscript and agree with its submission to MDPI Safety.
We look forward to hearing from you.
Sincerely,
Mairi Bowdler, Wouter Steijn & Dolf van der Beek
Mairi Bowdler (corresponding author)
TNO
Sylviusweg 71
2333 BE Leiden
e-mail: mairi.bowdler@tno.nl

Reviewer 3 Report
This Systematic-Review article has the following major errors that need to be corrected:
1) Figure 1 (p.4) needs to be modified to confirm that it is consistent with the content described in this paper.
2) Reference number 1 overlap (p.13 Lines 298-300) needs to be renumbered to be consistent with what is marked in the text of this paper.
3) Reference number 14 suggests that it can be omitted because "Skinner’s operant conditioning" may lead to unsafe behavior, such as the regulation to drive on the right or on the left.
4) On page 2, lines 69-71, the reference-cited in this review article description is not clear, please confirm again. (Reference number is 21, not 22)
No comments on the quality of english language.
Author Response
Manuscript re-submission Coverletter
Leiden, 20-06-2023
Dear Reviewer 3,
We would hereby like to resubmit the manuscript under the title “Effective components of behavioural interventions aiming to reduce injury within the workplace: a systematic-review.” We would like to thank the reviewer for providing us with the reviewer comments that have helped us improve on the manuscript. The exact nature of our changes and our response to the reviewer comments have been submitted elsewhere (please see the attachment).
We confirm that neither the manuscript nor any parts of its content are currently under consideration or published in another journal. All authors have approved the manuscript and agree with its submission to MDPI Safety.
We look forward to hearing from you.
Sincerely,
Mairi Bowdler, Wouter Steijn & Dolf van der Beek
Mairi Bowdler (corresponding author)
TNO
Sylviusweg 71
2333 BE Leiden
e-mail: mairi.bowdler@tno.nl

Reviewer 4 Report
The review paper focuses on organizational interventions aiming at occupational injury prevention and well falls within the scopes of the journal and the proposed SI.
The article is clearly written, English style is adequate and the textual elaboration is good.
As stated, the paper covers a topic well relevant to the scopes of the journal and recently attracting an increased research interest. Some revisions are suggested to increase the appeal for the reader. Introduction is not completely developed: it can be expanded to present a clear picture of international state-of-the-art covering the investigated topic, including relevant different scientific approaches. In addition, introduction needs to be enlarged to cover also industrial sectors at relevant risk where safety interventions are determining in reducing accident trend. As stated in Abstract, consideration is given to the area of "fatal and non-fatal injury within high-risk occupational industries", so authors should consider using as reference in introduction also the following articles which are fully embedded in the framework authors would analyze:
" A field study on human factor and safety performances in a downstream oil industry". DOI: 10.1016/j.ssci.2022.105795
"The cultural-historical development of occupational accidents and diseases prevention in France: A scoping review". DOI:10.1016/j.ssci.2022.106016
The discussion of results should be broadened in this regard. Likewise, in the discussion, reference clarify better possible limitations of the research and whether these limitations are more relevant in some specific industrial sector.
The choice of ROBIS tool in not fully justified and authors should consider also the possibility of different bias identification methods.
Finally, the conclusions should be expanded as a consequence of the recommendations indicated above for methodology and presentation and discussion of the results.
In Conclusion section the use of lumped reference should be avoided (note that one ref number is missing i.e. 55) for two reasons: a pertinent comment on each paper/contribution would be needed and, in the given case, it is considered sufficient referring to the first approach (49) and to latest developments (54).
The quality of English language is adequate.
Author Response
Manuscript re-submission Coverletter
Leiden, 20-06-2023
Dear Reviewer 4,
We would hereby like to resubmit the manuscript under the title “Effective components of behavioural interventions aiming to reduce injury within the workplace: a systematic-review.” We would like to thank the reviewer for providing us with the reviewer comments that have helped us improve on the manuscript. The exact nature of our changes and our response to the reviewer comments have been submitted elsewhere (please see the attachment).
We confirm that neither the manuscript nor any parts of its content are currently under consideration or published in another journal. All authors have approved the manuscript and agree with its submission to MDPI Safety.
We look forward to hearing from you.
Sincerely,
Mairi Bowdler, Wouter Steijn & Dolf van der Beek
Mairi Bowdler (corresponding author)
TNO
Sylviusweg 71
2333 BE Leiden
e-mail: mairi.bowdler@tno.nl

Round 2
Reviewer 1 Report
paper is ok
good
Author Response
We are delighted to observe that the reviewer has approved our revised manuscript. Following the suggestion, we engaged a native English speaker to thoroughly review the paper for any language-related concerns and implemented several enhancements.
Reviewer 3 Report
This revised file well done to accept.

Author Response
We are delighted to observe that the reviewer has approved our revised manuscript.
Reviewer 4 Report
Authors addressed the major part of reviewers' comments improving the overall appeal of the study.
Final minor English check to improve overall readability.
Author Response

(The authors gave the same response as above.)
